# Composite Coatings Based on Recombinant Spidroins and Peptides with Motifs of the Extracellular Matrix Proteins Enhance Neuronal Differentiation of Neural Precursor Cells Derived from Human Induced Pluripotent Stem Cells

**DOI:** 10.3390/ijms24054871

**Published:** 2023-03-02

**Authors:** Ekaterina V. Novosadova, Oleg V. Dolotov, Lyudmila V. Novosadova, Lubov I. Davydova, Konstantin V. Sidoruk, Elena L. Arsenyeva, Darya M. Shimchenko, Vladimir G. Debabov, Vladimir G. Bogush, Vyacheslav Z. Tarantul

**Affiliations:** 1Laboratory of Cell Differentiation, National Research Center “Kurchatov Institute”, 123182 Moscow, Russia; 2Laboratory of Molecular Neurogenetics and Innate Immunity, National Research Center “Kurchatov Institute”, 123182 Moscow, Russia; 3Faculty of Biology, Lomonosov Moscow State University, 119234 Moscow, Russia; 4Laboratory of Protein Engineering, National Research Center “Kurchatov Institute”, 123182 Moscow, Russia

**Keywords:** composite coatings, recombinant spidroins, fused peptides with ECM motifs, induced pluripotent stem cells, neural precursor cells, dopaminergic neurons

## Abstract

The production and transplantation of functionally active human neurons is a promising approach to cell therapy. Biocompatible and biodegradable matrices that effectively promote the growth and directed differentiation of neural precursor cells (NPCs) into the desired neuronal types are very important. The aim of this study was to evaluate the suitability of novel composite coatings (CCs) containing recombinant spidroins (RSs) rS1/9 and rS2/12 in combination with recombinant fused proteins (FP) carrying bioactive motifs (BAP) of the extracellular matrix (ECM) proteins for the growth of NPCs derived from human induced pluripotent stem cells (iPSC) and their differentiation into neurons. NPCs were produced by the directed differentiation of human iPSCs. The growth and differentiation of NPCs cultured on different CC variants were compared with a Matrigel (MG) coating using qPCR analysis, immunocytochemical staining, and ELISA. An investigation revealed that the use of CCs consisting of a mixture of two RSs and FPs with different peptide motifs of ECMs increased the efficiency of obtaining neurons differentiated from iPSCs compared to Matrigel. CC consisting of two RSs and FPs with Arg–Gly–Asp–Ser (RGDS) and heparin binding peptide (HBP) is the most effective for the support of NPCs and their neuronal differentiation.

## 1. Introduction

Tissue engineering aims to develop functional biological substitutes that restore, maintain, or improve broken tissue function by combining matrices, cells, and biologically active macromolecules. However, despite the notable successes of modern developments in the creation of various tissue engineering constructs, the task of obtaining matrices with optimized properties that improve the differentiation of NPCs into the desired types of neurons, the survival of transplanted neural cells and the high rate and degree of their vascularization and innervation required to ensure sufficient blood supply and normal functioning of the reconstructed nervous tissue still remain unsolved [1]. One of the attractive materials for such matrices are spidroins, which make up the frame filaments (dragline silk) of the spider web of orb weaving spiders. These proteins are characterized by unique mechanical properties—a combination of the highest values of strength and elasticity, which leads to high values of energy of rupture; they are also resistant to high and low temperatures and aggressive chemical influences [2]. Recombinant analogues of spidroins (RSs) have been obtained and their properties are largely close to natural [3]. Aside from high mechanical properties, RSs are biocompatible with any tissues of animals and humans, non-immunogenic, non-allergenic, and biodegradable to amino acids. Due to their self-assembly ability, RSs can form various supramolecular structures such as hydrogels including microgels, transparent elastic films, highly porous spongy 3D scaffolds, nonwoven materials, etc. [4]. In addition, the positive surface charge of these proteins and all devices based on them is very important for the adhesion of any cells including nerve cells to the substrate [5]. Another advantage of RS-based matrices is the possibility of introducing various biologically active peptides (BAPs) including motifs of the extracellular matrix (ECM) proteins into their structure. This allows obtaining matrices with unique predetermined properties, which are superior to synthetic and natural materials in terms of their breadth of application and the set of useful properties [6].

For tissue engineering, the sources of human nerve cells are also important. Significant progress in the technology of their production is associated with the creation of a reprogramming methodology, allowing for the transformation of the somatic cells of adult organisms into induced pluripotent stem cells (iPSCs) [7]. Various cocktails of small molecules and growth factors are used for directed neuronal differentiation of iPSCs, and the substrate on which this differentiation occurs is also extremely important. Currently, one of the most commonly used substrate is Matrigel (MG), derived from the basal membrane of Engelbreth–Holm–Swarm mouse sarcoma, which is rich in various ECM proteins including laminin, collagen IV, proteoglycans, heparin sulfates, and growth factors. However, this matrix cannot be used for regenerative medicine due to its oncogenic origin and potential for pathogen contamination [8].

Among the currently used matrix materials, RSs appear to be among the most promising for dealing with neuronal differentiation of cells derived from iPSCs in vitro. Some RSs have been shown to be effective agents in brain repairing after stroke due to the presence of multiple repeats of the GRGGL sequence recognized by NPC [9]. RS-based matrices support the proliferation and neuronal differentiation of neural stem cells. They have a more suitable surface charge (positive over the entire range of physiological pH values) and stiffness to support NSC growth than matrices made from silk fibroin or polylysine [10]. Previously, it has also been shown that the nonwoven matrix based on RSs rS1/9 and rS2/12, polycaprolactone and platelet-rich plasma, supports growth and neuronal differentiation of human NPCs [11]. 

The aim of this study was to evaluate the suitability of novel CCs containing RSs rS1/9 and rS2/12 in combination with the first time obtained by genetic engineering fused proteins (FPs) containing SUMO, RS rS1/9 “monomer”, and some BAP of the ECM proteins, for the growth of NPCs derived from human iPSCs and their differentiation into dopaminergic (DA) neurons.

## 2. Results

### 2.1. The Influence of Different CC Variants on the Proliferative Activity of NPCs Differentiating from iPSCs

The following CCs were used in this work:

SP1: rS2/12 + FP(RGDS);

SP2: rS2/12 (control for SP1 and SP5);

SP3: rS1/9 + rS2/12 + FP(RGDS) + FP(HBP);

SP4: rS1/9 + rS2/12 (control for SP3);

SP5: rS2/12 + FP(GRGGL),

where FP(RGDS) is the fused protein with the RGDS motif; FP(HBP) is the fused protein with the heparin binding peptide (HBP) motif; and FP(GRGGL) is the fused protein with the GRGGL motif. The Matrigel coating (MG) was used as the positive control.

In the first stage of the work, we evaluated the ability of five different variants of CCs based on RSs rS1/9 + rS2/12, both individually and in combination with FP, to support the growth and proliferation of NPCs obtained by the directed differentiation of human induced pluripotent cells (iPSCs). The MG coating served as a comparison. A schematic of the experiment is shown in Figure 1.

Nerve cells derived from iPSCs at different stages of differentiation (NPC, IDN, and DN) were initially characterized using qPCR by the level of transcription of early and late neuronal markers in them. It was shown that as neuronal differentiation proceeds, the expression level of early neuronal markers (*NESTIN*, *PAX6*, *SOX2*) decreased in the cells, while the expression level of late ones (*TUBB3*), in contrast, increased (Figure 2).

Next, the ability of NPCs and IDNs to proliferate when cultured on CCs compared with the MG coating was assessed (Figure 3).

As can be seen from the histograms in Figure 3a, the proliferation of NPCs is reduced when cultured on all CCs compared to MG with the exception of SP3. At the same time, the observed decrease in IDN proliferative activity when cultured on SP3 was minimal compared to the other variants including MG. The adhesive properties of most of the studied CCs for NPS and IDN were also decreased compared to MG, the only exception being SP3 (Figure 3).

### 2.2. Study of the Effect of Cultivation on CCs on the Differentiation of NPS into IDNs

To evaluate the effect of different CC variants on NPC differentiation, the expression levels of mRNA specific for early (*PAX6*, *SOX2* and *NESTIN*) and late (*TUBB3*) neuronal genes were analyzed in these cells using qPCR. NPCs were dispersed onto coating-treated culture dishes and cultured for 5 days. We found that the expression level of early and late neuronal marker genes did not differ significantly by cultivation NPCs on all of the analyzed coatings (Table 1).

Analysis of early neuronal marker expression at the protein level using immunocytochemical staining showed that the NPC stage contained more than 75% SOX2-positive cells in the population (Figure 4). The number of such cells was not statistically different when cultured on different CCs and MG, which confirms the data from the qPCR analysis (Figure 4a). Given the fact that NPCs can differentiate in both the neuronal and glial direction, we estimated the number of spontaneously differentiated glial cells and showed that they represented less than 1% of the total population (Figure 4a). Figure 4b shows the representative photos of the immunocytochemical staining of the resulting cell population.

Next, we analyzed the effect of cultivation on different CCs on IDN differentiation from the NPCs. For this purpose, the NeuN protein, which is a nuclear protein present in postmitotic neurons, was used as a marker protein. We found that the number of NeuN-positive cells did not differ between different cells up to 41 days of cultivation (Figure 5).

### 2.3. Influence of Cell Cultivation on CCs on Differentiation of NPCs into DA Neurons

The dopaminergic (DA) neurons play an essential role in maintaining the human brain’s normal sensation, voluntary movement, emotion, and cognition [12]. The previously described protocol was used for the directed production of DA neurons from NPCs [13]. In IDN and DN obtained by the cultivation of NPS on different CCs, we performed a comparative study of the mRNA expression levels of the genes’ characteristic of DA neurons—tyrosine hydroxylase (*TH*) and the aromatic L-amino acid decarboxylase (*AADC*) (Figure 6).

As can be seen from Figure 6, there was a significant increase in the expression of *TH* and *AADC* genes specific to DA neurons in DNs formed by cultivation on different CCs. This suggests that these CCs are very effective in affecting differentiation when using our protocol of the directed differentiation of NPCs into DA neurons.

Using immunocytochemical analysis with anti-TH antibodies, we showed that in DNs, despite a significant increase in *TH* gene transcription, this change was less significant at the protein level (about 40%) and was observed only when cells were cultivated on the SP3 and SP5 coatings (Figure 7).

Figure 8 shows the representative photographs of DN cultured on different CCs.

### 2.4. Influence of Cell Cultivation on CCs on Synaptogenesis in Emerging NPCs and IDNs

The study of the transcription of marker genes responsible for synaptogenesis showed that the expression of the *SNAP25*, *STX1A*, *SNPT*, *SYN2*, and *SYN3* genes did not differ significantly from all the CCs and MG used for the cultivation NPCs (Table 2).

Further differentiation of NPCs into IDNs when cultivated on all CCs (except SP4) resulted in the increased expression of the synapse-specific genes *SNAP25*, *STX1A*, *SYN2*, and *GSG1L* compared to MG. By day 45 of differentiation, the expression levels of the studied genes in DN generally leveled off. There was only an increase in *SNPT* expression in DNs when they were cultivated on SP1 and SP5 (Table 3).

Synapsins (*SYN*, *SYN2*, and *SYN3*) are also important markers of cellular synaptogenesis ability. As can be seen from Table 4, only IDNs showed a marked increase in the transcription of individual genes of this family when cells were cultivated on CCs.

Thus, we can conclude that the studied coatings based on RSs and FPs compared to MG contribute to the enhancement of synaptogenesis in the process of neuronal differentiation of NPS into IDN. The main difference in the expression of genes involved in synaptogenesis mainly occurs with coatings with BAP (SP1, SP3 and SP5).

### 2.5. Effect of FPs on the Efficiency of Synaptogenesis during Neuronal Differentiation on CCs

To establish the role in synaptogenesis of BAPs included in SP1, SP3, and SP5 coatings (RGDS, RGDS + HBP and GRGGL motifs, respectively), as part of FP, the effect of these CCs was compared with that of the CCs without BAPs (SP2 and SP4). SP2 consisting only of rS2/12 was used as the internal controls for SP1 and SP5, and SP4 coatings consisting only of a mixture of two RSs (rS1/9 + rS2/12) were used as a control for SP3. For this purpose, we compared the expression levels of genes involved in synaptogenesis in the IDNs and DNs obtained by cultivating on different controls with their internal controls. Table 5 and Table 6 present the data of the qPCR analysis of the transcription of various genes involved in synaptogenesis as ratios of the gene expression levels in IDN and DN cultured on the coating with added FPs (SP1, SP3, SP5) to SP2 and SP4 not containing these proteins.

In the intermediate stage of neuronal differentiation (IDN), all coatings with BAP showed an increased expression of the synaptogenesis marker genes *GSG1L* and *SYN2*. The expression in the *STX1A* gene was enhanced in IDNs obtained when the cells were cultured on SP5, and the expression of the *SNAP25* gene was enhanced in IDNs obtained on the SP3 coating. At the same time, the maximum difference in expression was observed with the SP3 coating compared with MG (Table 6).

Thus, we can conclude that the presence of BAP in the coatings contributes to the enhancement in the expression of a number of genes involved in synaptogenesis in IDNs. The maximum activation of the transcription of these genes was observed when cells were cultured on the SP3 coating containing BAPs both with RGDS and HBP motifs.

### 2.6. Effect of Cultivation on CCs on the Expression of Neurotrophic Factors (NTF) Genes in IDNs and DNs Formed on Them

Next, we investigated the effect of the coatings analyzed on the expression in the IDNs and DNs of neurotrophic factor (NTF) genes, which are necessary for normal differentiation and the maintenance of neuronal viability. When cultured on all coatings in the IDN intermediate stage, there was a significant increase in the expression of the *BDNF*, *GDNF*, and *NGF* genes, while in contrast, the *NT3* gene decreased its expression (Table 7). At the same time, in DNs cultured on CCs, the NTF expression levels were commensurate with the cells cultured on MG.

The protein amounts of BDNF and GDNF were assessed in DNs differentiated on different CCs using commercial enzyme-linked immunosorbent assay (ELISA) kit. It was found that when cells were cultured on the SP1, SP3 and SP5 coatings, the levels of both BDNF and GDNF in the cell lysates were many times (more than 10-fold) higher than when MG was used as a coating (Figure 9). At the same time, BDNF and GDNF were not detected in the conditioned media collected from the corresponding cultures.

Since the SP1, SP3, and SP5 coatings contain FPs with BAPs, the results indicate that peptides of ECM proteins contribute to the enhancement of NTF synthesis in CC-forming DNs.

## 3. Discussion

Cell therapy for neurodegenerative diseases requires the creation of functionally active neuronal constructs that can be used for transplantation. This goal, first of all, requires the selection of NPCs as well as the selection of matrices that allow these cells to grow and differentiate into mature neurons. The task of the present study was to find the optimal composition of RS-based matrices that would most effectively promote the growth and directed differentiation of human iPSC-derived NPCs into the desired neuronal types and provide increased survival of transplanted nerve cells after their transplantation.

Our previous experience of using RSs in the form of microgels, highly porous spongy like 3D scaffolds, isotropic and anisotropic nonwoven matrices indicates that these materials are effective for neuronal cell growth and differentiation and can induce neoangiogenesis and neoinnervation when transplanted into the lesion area in the animal body, which is crucial for damaged tissue regeneration.

The choice of RSs as a base material for such matrices is associated with the previously obtained results of successful applications of these proteins for the cultivation of various types of cells and in experiments on laboratory animals. For example, a layer of isolated neonatal rat cardiomyocytes was grown on a nonwoven matrix of RS rS1/9, rS2/12, and rS2/12-RGDS obtained by electrospinning and not containing any additional biologically active compounds. Optical excitation mapping proved that the cells do indeed form syncytium, and the excitation impulse travels through the grown tissue, causing synchronous cell contraction, as in in vivo cardiac tissue [14,15].

In experiments on laboratory animals, it was found that bioengineered microparticles from RS perform not only the function of a framework for cells, but are themselves capable of influencing the immune response and have pro-regenerative properties [16]. It was also shown that rS1/9-based anisotropic nonwoven matrices in combination with platelet-rich plasma are a suitable biocompatible substrate for reprogrammed NPCs when implanted into the brain and spinal cord of rhesus macaques [11]. BAPs with ECM motifs have long been utilized in the creation of scaffolds for neural tissue cells as components of matrices [17].

Cultivation of NPCs on anisotropic matrices based on rS1/9 PC and BAPs with motifs from ECM proteins (RGD from fibronectin, IKVAV laminin pentapeptide, and VAEIDGIEL motif from tenascin-C) mainly preserved their stemness in the growth medium [18]. It was demonstrated that different motifs have different effects on neurogenesis: the RGD motif promotes the formation of a smaller number of neurons with longer neurites, whereas the IKVAV motif is characterized by the formation of more NF200-positive neurons with shorter neurites. In experiments with nonwoven matrices made of RS with oriented fibers, they have been shown to direct the migration of Schwann cells and accelerate axonal growth from mouse dorsal ganglia as well as induce the migration of smooth muscle and aortic endothelial cells [19].

The present work is a logical continuation of our series of studies on the effect of RS-based matrices and their derivatives on the growth and differentiation of human and animal nerve cells in vitro and in vivo. In contrast to previous studies, the object of the present work was NPCs derived from human iPSCs.

To study the growth and differentiation of NPS in vitro, a mixture of previously genetically engineered RS (rS1/9 and rS2/12) and FPs with three BAPs was used as a coating: RGDS tetrapeptide from fibronectin that recognizes the integrins of most cells; the GRGGL pentapeptide, which is recognized by NCAM and provides good adhesion of neural precursors, and heparin-binding peptide (HBP). HBP is part of laminin, which is a major component of the basal membrane surrounding the brain and blood vessels throughout the CNS [20], and is also present in the ventricular zone of the developing neocortex. Laminins have been shown to promote the expansion, migration, and differentiation of NSCs in vitro [21,22]. In addition, HBPs have been found to be involved in the binding of various growth factors and to interact with syndecans in the cell membrane [23].

Cultivating NPS and NDN on most of the studied CCs revealed a decrease in the adhesive properties and ability to support cell proliferation compared to MG, the only exception being SP3 coatings (Figure 3). At the same time, we found no change in the expression levels of early and late neuronal marker genes in cultivated NPCs on all the analyzed coatings (Table 1). However, a further comparison of the effect on the growth and differentiation of NPCs when they were cultured on CCs and standardly used MG, showed a marked advantage of the former. It was shown that in the DA neurons formed on CCs, there was a multiple increase in the expression of the *AADC* and *TH* genes characteristic of this cell type (Figure 6).

These data were partially confirmed by immunocytochemical staining with antibodies to tyrosine hydroxylase: in particular, there was an increase in the number of DA neurons on the SP3 and SP5 coatings (Figure 7). Thus, we can conclude that these CCs are more effective than the MG coating for the directed differentiation of NPCs into DA neurons.

In addition, IDNs cultured on CCs (Table 4 and Table 5) showed a marked increase in the transcription of certain genes involved in synthapogenesis compared to cells cultured on the MG coating (Table 3 and Table 4). This may indirectly indicate the acceleration of neuronal differentiation on the studied CCs. The maximum number of genes that changed their expression at this differentiation stage was observed when the cells were cultured on CCs SP1, SP3, and SP5. At the stage of DN, an increase in the expression of the *SNPT* gene was noted when cells were cultured on SP1 and SP5. To assess the role in the synaptogenesis of BAPs included in the FPs of the SP1, SP3, and SP5 coatings, the effects of these coatings were compared with the effects of CCs without BAPs (SP2 and SP4). It was found that all BAPs, to varying degrees, caused the upregulation of the studied genes. The maximum number of upregulated genes (*SNAP25*, *SNPT*, *PSD95*, *SYN2*, and *GSG1L*) was observed for SP3. The cultivation of cells on SP5 led to increased expression of *STX2*, *SYN2*, and *GSG1L*, and on SP1, only *SYN2* and *GSG1L*.

The secretion of NTFs, secretory dimeric proteins that have a significant influence on all biological processes of neurons during pre- and postnatal ontogenesis, is essential for the normal development and viability of neurons. In the developing nervous system, neurotrophins regulate cell division, cell migration, differentiation, establishment, and maintenance of intercellular contact activity as well as the initiation of apoptosis [24,25,26].

It was shown that the expression of most of the studied NTF genes (*BDNF*, *GDNF*, *NGF*) was enhanced at the NDN stage, the only exception being *NT3*, whose expression, in contrast, was decreased in all CC variants compared to the MG coating (Table 7). The maximum difference was observed for the SP1, SP3, and SP5 coatings, while for these same CCs, the increased expression of these genes persisted in DN, which was confirmed for BDNF and GDNF at the protein level (Figure 9).

The fact that the increased expression of most of the studied genes involved in neuronal differentiation was observed for the SP5 variant containing the GRGGL sequence, in contrast to the SP2 variant that does not contain this sequence, which indicates the effect of GRGGL on the differentiation process. At the same time, a significant increase in the expression of the studied genes was found when the cells were cultivated on the SP5 coating compared to SP4 containing the rS1/9 protein, which includes 18 repeats of the GRGGL sequence [27]. This can be explained by the lower availability of GRGGL for contact with NCAM in the cell walls compared to the same in SP5, where it was exposed above the surface of the coating due to the presence of a 14-mer linker (SGG)4S, which ensures the binding of this peptide to the rest of FP and gives it extra mobility.

The found influence of various BAPs on the differentiation of neural progenitors through the activation of the expression of the studied genes normally involved in differentiation is associated with the known role of these peptides in cell activation. This activation is mediated via various pathways of interaction of the BAPs with cells: RGDS interacts with integrins, BAP interacts with syndecans, GRGGL interacts with NCAM. A positive effect on the expression of genes involved in neuronal differentiation was already observed for individual BAPs (RGDS and GRGGL in SP1 and SP5, respectively). At the same time, the maximum effect, as expected, was found for the SP5 coating containing all three BAPs used in the work (GRGGL in the rS1/9 protein, RGDS, and HBP). These results are in good agreement with the known literature examples of the use of similar BAPs for the adhesion, proliferation, and differentiation of neuronal cells [28,29,30].

Thus, certain newly created RS- and FP-based CCs with ESM motifs promote NPC differentiation into DN by enhancing the expression of genes involved in synaptogenesis, stimulating the synthesis of a number of NTFs and contributing to the production of human DA neurons.

## 4. Methods and Materials

### 4.1. Isolation and Purification of Full Size RSs

We used two RS—rS1/9 and rS2/12, whose genes we previously cloned in *Saccharomices cerevisiae* [14,27]. The rS1/9 molecule has a molecular mass of 94 kDa and consists of nine so called monomers, consisting of four initial repeats. Each of them contains GGX tripeptides (X = L, Y, Q) and one poly-Ala cluster consisting of five to eight Ala residues. This protein also contains 18 repeats of the NCAM-binding sequence GRGGL, which is a signal for binding neuronal cells. The rS2/12 molecule has a molecular mass of 113 kDa and consists of 12 so called monomers, consisting of five initial repeats. Each repeat contains pentapeptides GPGGY and GPGQQ and also one poly-Ala cluster consisting of five to eight Ala residues. These poly-Ala clusters form β-sheets, which, in turn, form crystallites that provide a unique stability to the materials based on spidroins [27]. Yeast biomass production, RS isolation, and purification by ion-exchange chromatography using a HiPrep 16/10 SP FF column (GE Healthcare, Chicago, IL, USA) and an ACTA purifier TM chromatograph (GE Healthcare, Chicago, IL, USA) with pH exchange (pH 4.0–pH7.0–pH4.0) were carried out in accordance with previously published protocols [18]. The RSs were eluted from the column and dialyzed against deionized water and then frozen and lyophilized as described.

### 4.2. Obtaining and Purification Fused Peptides (FPs)

FPs containing different BAPs were designed according to the same scheme: H_6_-SUMO-rS1/1-G(SGG)_4_S-[BAP], where H_6_ is the his-tag for ease purification of FPs on a Ni-column; SUMO (Small Ubiquitin Like Modifier) is a peptide product of the yeast Smt3 gene [31], which, according to the literature [32] and our experience, increases the yield of the product in *E.coli* cells (can dramatically improve protein solubility, achieve native protein folding, and increase total yield by improving expression and decreasing degradation); rS1/1—monomer of rS1/9, which acts as an “anchor” in the interaction with full-sized RSs; G(SGG)_4_S is a neutral linker that promotes the exposure of biologically active peptide (BAP) over the matrix surface; BAP—any of the cloned BAP.

We chose the following polypeptides as BAPs: RGDS, a tetrapeptide from fibronectin that recognizes the integrins of most cells [33,34]. GRGGL is a pentapeptide that is recognized by NCAM, interacts with neuron surface receptors, and upregulates NCAM expression in primary cortical neurons from embryonic day 18 (E18) Sprague–Dawley rats [9]; HBP—heparin binding peptide (GGGGSPPRRARVTY) [35], which is involved in the binding of various growth factors and interacts with syndecans in the cell membrane [23].

The genes of all three FPs were designed and chemically synthesized. The codons in the sequences were optimized to facilitate the synthesis of the construct: the rarest codons in E. coli were removed. The resulting constructs were cloned in the pET-28a-Novagen expressive vector (Novagen, Merck KGaA, Darmstadt, Germany) at the NcoI and XhoI restriction sites and transformed into the *E. coli* strain BL21(DE3) (Novagen, Merck KGaA, Darmstadt, Germany)) using the same vector. As a result, three strains, producing FPs: FP(RGDS), FP(GRGGL), and FP(HBP) were obtained.

To isolate FPs, the strain producers were grown in a 10 L fermenter in a growth medium (20 g/L soy peptone Amresco 140 (VWR Life Science AMRESCO, Cambridge, MA, USA); 10 g/L yeast extract Maisons-Alfort, France; 5 g/L glucose (Acros Organics, Waltham, MA, USA); 5 g/L NaCl (Scharlab, Sentmenat, Barcelona, Spain); 0.5 g/L kanamycin (VWR Life Science AMRESCO, Radnor, PA, USA)); up to stationary phase; induction was carried out with lactose as part of the feed (20 g/L soy peptone Amresco 140; 10 g/L yeast extract; 30 g/L glucose; 5 g/L NaCl; 0.5 g/L kanamycin). The process was carried out at 28 °C, pH 7.0 with a typical growth time of 16–18 h.

Biomass was harvested by centrifugation (14,000× *g* at 4 °C for 30 min) and suspended in buffer (0.05 M Na-Pi buffer, pH 8.0; 0.2 M NaCl; 5% glycerin; 0.02 M imidazol) at a ratio of 9:1. Suspension was sonicated in 50 mL of buffer and clarified by centrifugation.

To purify the FP, chromatography with the FPLC system ÄCTApurifier^TM^ an ACTA (GE Healthcare, Chicago, IL, USA) with an installed HiTrap 5 mL of the NiNTA resin column (GE Healthcare, Chicago, IL, USA) was utilized. The elution was carried out stepwise using a decrease in pH to 6.0 and an increase in the content of imidazole (BioFroxx, Bruckberg, Germany) to 0.45 M in a buffer of the same composition. The desired proteins were detected using electrophoresis in 15% PAAG-SDS. The proteins after chromatographic purification were dialyzed against deionized water, frozen at −70 °C and freeze-dried. Protein concentrations were determined by spectrometry at 280 nm. After purification, the samples contained >95% of the target protein.

### 4.3. Preparation of Mixed Protein Solutions for Dishes Coating

Freeze-dried proteins (both full-length RSs and FPs) were dissolved in concentrated (99.7%) formic acid (Helicon, RF) to a final concentration of 400 mg/mL for 14–16 h until complete dissolution. Then, the protein solutions were mixed in such a way that the final total concentration of all proteins was equal to 400 mg/mL, while the total concentration of FP in each solution was 10% of the total protein. After that, each sample was diluted 100 times with deionized water. The final total protein concentration for all samples was 2 mg/mL, and the concentration of formic acid was 1%. Solutions were centrifuged at 18,000× *g* at 4 °C for 30 min to remove protein aggregates immediately prior to use.

In the preliminary experiments, it was found that the optimal ratio between the full-size RSs and FPs was a ratio of 9:1 (by weight), so this ratio was used in all experiments. This meant that the total mass of full-sized RSs in the mixture was always 90%, and the total mass of FPs was 10%. At the same time, both rS1/9 and rS2/12 among themselves, and FP among themselves had always been in an equal ratio. Coating RS and FP mixtures were prepared immediately prior to use. To do this, we mixed the prepared solutions of each protein in the selected ratio.

MG solution was prepared according to the manufacturer’s protocols (Corning Life Sciences, NY, USA).

In this work, six variants of samples for the coating cups were used:

SP1: rS2/12 + FP(RGDS) in relation to 9:1;

SP2: rS2/12 (control for SP1 and SP5);

SP3: rS1/9 + rS2/12 + FP(RGDS) + FP(HBP) in relation to 4.5:4.5:0.5:0.5;

SP4: rS1/9 + rS2/12 (control for SP3) in relation to 5:5;

SP5: rS2/12 + FP(GRGGL) in relation to 9:1;

MG: (Matrigel, positive control).

where FP(RGDS) is the fused protein with the RGDS motif; FP(HBP) is the fused protein with the HBP motif; and FP(GRGGL) is the fused protein with the GRGGL motif.

### 4.4. The Coating Preparation

The coating was carried out as follows: 1 mL of the prepared protein solutions was poured into a sterile Petri dish (d = 35 mm) and incubated in a laminar box for 30 min, the protein solution was removed and to stabilize the coatings, the samples were immersed in 96% (*v/v*) ethanol for 30 min to induce a β-sheet structure [36]. After that, the dishes were incubated for 30 min in sterile deionized water followed by 70% ethanol for 30 min and then in sterile deionized water. This procedure was repeated 10 more times. The cups were dried and used immediately, or wrapped with Parafilm and stored at +4 °C for a month.

A standard cup coating procedure was used for the Matrigel. A total of 1 mL of the solution was poured into a sterile Petri dish (d = 35 mm) and incubated for 60 min. The treated cups were used immediately or wrapped with Parafilm and stored at +4 °C for a month. Immediately before use, the Matrigel was removed and washed once with DMEM medium containing penicillin–streptomycin (50 U/mL; 50 µg/mL) (Paneco, Moscow, Russian Federation).

### 4.5. Ethics Statement

The study complies with the World Medical Assembly Declaration of Helsinki—Ethical Principles for Medical Research Involving Human Subjects. This work was approved by the Ethic Committee of the Institute of Molecular Genetics of National Research Centre “Kurchatov Institute” (Protocol no. 3 from 19 February 2018). Donor provided a written informed consent.

### 4.6. Human Pluripotent Stem Cell Culture

The work was carried out on the iPSC line (IPSHD1.1S) obtained from skin fibroblasts of a healthy donor using the CytoTune™-iPS 2.0 Sendai Reprogramming Kit. The reprogramming vectors included the four Yamanaka factors, Oct, Sox2, Klf4, and c-Myc, shown to be sufficient for efficient reprogramming. The obtained iPSC expressed the essential pattern of specific pluripotency-associated genes, possessed a normal karyotype, and were capable of producing the derivatives of embryonic threenic germ layers [37]. Cells were cultured in StemMACS iPS-BrewXF medium (Miltenyi Biotec, Nordrhein-Westfalen, Germany) on Matrigel (Corning Life Sciences, NY, USA) treated with Petri dishes. The medium was changed daily.

### 4.7. Generation of Human iPSC-Derived Neural Stem Cells (NSCs)

iPSCs were cultured in CO_2_ incubator (5% CO_2_, 80% humidity and 37 °C) in iPS-Brew XF basal medium (Miltenyi Biotec, Nordrhein-Westfalen, Germany) until reaching an 80% confluent monolayer. The culture medium was replaced by the medium for neural progenitors. After 10–14 days of cultivation, neural rosettes with specific “ridges” were formed. Rosettes were mechanically transferred to a 24-well plate with ultra-low adhesion (Corning Life Sciences, NY, USA) and cultivated for 3–5 days until neurospheres were formed. Neurospheres were collected and treated with 0.05% trypsin (ICN Biomedicals, Hackensack, NJ, USA). After trypsin inactivation in DMEM supplemented with 10% FBS (HyClone, Waltman, MA, USA), cells were resuspended in growth medium for neural progenitors with 5 µM Rock (StemoleculeY27632, Stemgent, Cambridge, MA, USA), and transferred to Petri dishes coated with Matrigel (Corning Life Sciences, NY, USA). NPs were cultivated to a dense monolayer, changing the medium every 48 h. After reaching the monolayer, NPCs were plated with 0.05% trypsin on new Petri dishes coated with Matrigel at a dilution of 1:4 or 1:5. Cells were cultured in a CO_2_ incubator (5% CO_2_, 80% humidity, and 37 °C).

To study the effect of different matrices on the proliferation and differentiation of NPCs, cells of 2–3 passages were used. NPCs were dispersed onto Petri dishes pretreated with MG and RS.

### 4.8. Media Used for Cultivation and Differentiation of Neurons

Culture medium for NPCs:

Neurobasal medium (Gibco, Carlsbad, CA, USA), penicillin–streptomycin (50 U/mL; 50 µg/mL) (Paneco, Moscow, Russian Federation), 2% serum replacement (Gibco, Carlsbad, CA, USA), 1% N2 (Life Technologies, Carlsbad, CA, USA), 2 mM L-glutamine (ICN Biomedicals Inc., Hackensack, NJ, USA), 1 mM non-essential amino acids (Paneco, Moscow, Russian Federation), 10 μM SB431542 (Stemgent, Cambridge, MA, USA), and 80 ng/mL recombinant Noggin (Peprotech, Cranbury, NJ, USA).

Culture medium for neuronal differentiation type I (NDN):

Neurobasal medium A, (Gibco, Carlsbad, CA, USA), penicillin–streptomycin (50 U/mL; 50 µg/mL) (Paneco, Moscow, Russian Federation), 2% serum replacement (Gibco, Carlsbad, CA, USA), 1% B-27 (Life Technologies, Carlsbad, CA, USA), 2 mM L-glutamine (ICN Biomedicals Inc., Hackensack, NJ USA), 1 mM non-essential amino acids (Paneco, Moscow, Russian Federation), 100 ng/mL human SHH (Miltenyi Biotec, Nordrhein-Westfalen, Germany), 100 ng/mL FGF8 (PeproTech, Cranbury, NJ, USA), 10 μM purmorphamine (Sigma-Aldrich, Saint Louis, MO, USA).

Culture medium for neuronal differentiation type II (DN):

Neurobasal medium A, (Gibco, Carlsbad, CA, USA), penicillin–streptomycin (50 U/mL; 50 µg/mL) (Paneco, Moscow, Russian Federation), 2% serum replacement (Gibco, Carlsbad, CA, USA), 1% B-27 (Life Technologies, Carlsbad, CA, USA), 2 mM L-glutamine (ICN Biomedicals Inc, Hackensack, NJ, USA), 1 mM non-essential amino acids (Paneco, Moscow, Russian Federation), 20 ng/mL BDNF (PeproTech, Cranbury, NJ, USA), 20 ng/mL GDNF (PeproTech, Cranbury, NJ, USA), 200 μM ascorbic acid (StemCell, Vancouver, BC, USA), 4 μM Forskolin (Stemgent, Cambridge, MA, USA).

### 4.9. Targeted Differentiation of NPCs in IDN and DN

The NPs were disseminated at 200,000 cells per cm² into Petri dishes pre-treated with MG and RS in neuronal precursor medium supplemented with 5 µM Rock (StemoleculeY27632, Stemgent, Cambridge, MA, USA). The next day, the medium was replaced with medium for the differentiation of type I neurons. The cells were cultured for 10 days, with the medium changing every other day. After the cells reached a dense monolayer, they were disseminated to new 1:4 or 1:5 cups. On the ninth day of cultivation, the cells were removed from the substrate with 0.05% trypsin (Gibco, Carlsbad, CA, USA) and disseminated on a prepared culture dish at 400,000 cells per cm^2^ in medium for the differentiation of type I neurons with the addition of 5 µM Rock (StemoleculeY27632, Stemgent, Cambridge, MA, USA). The next day, the medium was changed to medium for the differentiation of type II neurons and the cells were cultured for 14 days. The medium was changed every other day for the first 7 days and daily thereafter. Cells were cultured in CO_2_-incubator (5% CO_2_, 80% humidity, and 37 °C).

### 4.10. Immunocytochemistry

The adherent cells on the Petri dish were washed with PBS, fixed with 4% para-formaldehyde in PBS (pH 6.8) for 20 min at room temperature (RT), and washed in PBS with 0.1% Tween 20 (Sigma-Aldrich, Saint Louis, MO, USA) three times for 5 min. Nonspecific antibody sorption was blocked by incubation in blocking buffer (PBS with 0.1%, Triton x100, and 5% fetal bovine serum (HyClone, Waltman, MA, USA)) for 30 min at RT. Primary antibodies (Table 8) were applied overnight at 4 °C, and then washed in PBS with 0.1% Tween 20 three times for 5 min. The secondary antibodies were applied for 60 min at RT, then washed in PBS with 0.1% Tween 20 three times for 5 min. After that, the cell cultures were incubated with 0.1 μg/mL DAPI (Sigma-Aldrich, Saint Louis, MO, USA) in PBS for 10 min for visualization of the cell nuclei, and washed twice with PBS. The cells were investigated using an AxioImager Z1 fluorescence microscope (Carl Zeiss, Oberhohen, Germany), and images were taken with AxioVision 4.8 software (Carl Zeiss, Oberhohen, Germany). For cell counting, the multiple fields that covered the whole dish surface were imaged. The obtained images were analyzed with ImageJ 1.49 software (NCBI, Bethesda, MD, USA) using the ITCN plugin (Center for Bio-image Informatics, Santa Barbara, CA, USA).

### 4.11. Quantification of BDNF and GDNF Protein Levels in Cultured Cells

The levels of BDNF and GDNF proteins were quantified using a sandwich ELISA. Culture media were collected at indicated time points, centrifuged at 14,000× *g* for 5 min at 4 °C, and the supernatants were stored at −80 °C. The cells were washed three times with cold PBS before being lysed in 1 mL of 100 mM PIPES lysis buffer, pH 7.0, containing 500 mM NaCl, 2% BSA, 0.2% Triton X-100, 0.1% NaN3, and protease inhibitors (2 μg/mL aprotinin, 2 mM EDTA, 10 μM leupeptin, 1 μM pepstatin, and 200 μM PMSF) [38]. The sister wells were treated with trypsin and viable cells were counted using trypan blue exclusion and a hemocytometer. After three freeze/thaw cycles, the cell lysates were centrifuged at 14,000× g for 5 min at 4 °C, and the supernatants were stored at −80 °C. The BDNF and GDNF concentrations in the cell supernatants and lysates were determined in duplicate using Human/Mouse BDNF DuoSet ELISA and Human GDNF DuoSet ELISA Kits (R&D Systems, Minneapolis, MN, USA), according to the manufacturer’s instructions.

### 4.12. Quantitative Real-Time PCR (qPCR)

Total RNA was extracted from the cells with a Trizol RNA Purification Kit (Invitro-gen, USA) following the manufacturer’s instructions, with a subsequent DNA-Free DNA Removal Kit (Invitrogen, Carlsbad, CA, USA) treatment. cDNA was synthesized on 0.5–2 μg of total RNA using M-MLV Reverse Transcriptase (Evrogen, Moscow, Russian Federation) with random primers. The primer sequences are shown in Table 9. The cDNA obtained was amplified using a CFX96 Touch™ Real-Time PCR Detection System (Bio-Rad, Berkeley, CA, USA) set to the following reaction conditions: denaturation at 95 °C (3 min), cycles n = 40 (95 °C, 15 s; 60 °C, 20 s; 72 °C, 45 s). The qPCRmix-HS SYBR reaction mixture (Evrogen, Moscow, Russian Federation) was used and 18S rRNA was accepted as the reference gene.

### 4.13. MTT (3-(4,5-dimethylthiazol-2-yl)-2,5-diphenyltetrazolium bromide) Assay

For the quantification of cell growth and viability, adhesive cell cultures were incubated for 4 h in culture medium containing 0.5 mg/mL MTT (Sigma-Aldrich, Saint Louis, MO, USA). Next, the medium was removed and the blue MTT–formazan product was diluted with DMSO (Panreac, Barcelona, Spain). After 2 h of incubation at RT on a shaker setting of 150 rpm/min, the absorbance of the formazan solution was recorded at 600 nm using a spectrophotometer (Metertech, Taiwan).

### 4.14. Statistical Analyses

Data were analyzed using GraphPad Prism software. Normality and homogeneity of variance were assessed by Shapiro–Wilk and Brown–Forsythe tests, respectively. Statistical analyses of the data were performed using the Student’s t-test or ordinary one-way ANOVA followed by Dunnett’s post hoc test wherever applicable, as indicated in the figures and tables legend. Results are presented as the mean ± SEM.

## Figures and Tables

**Figure 1 ijms-24-04871-f001:**
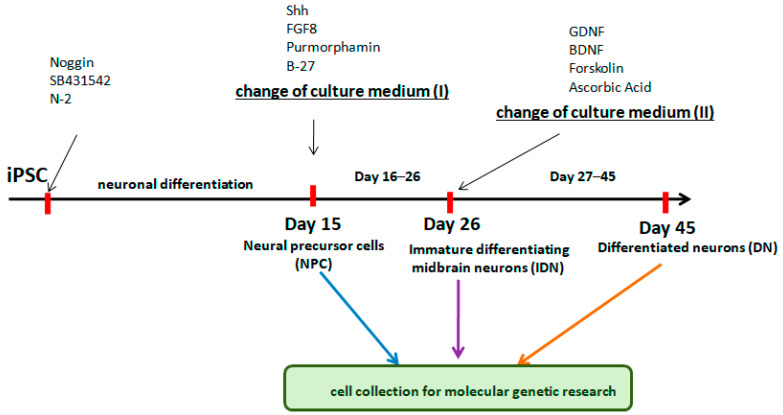
Schematic flow diagram to depict the stages of differentiation into neurons from iPSCs.

**Figure 2 ijms-24-04871-f002:**
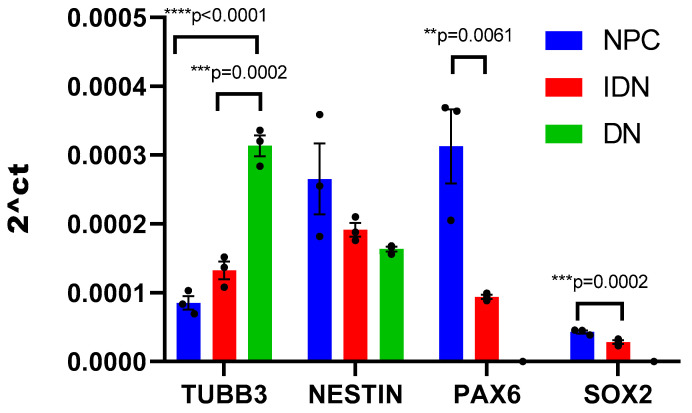
Neuron-specific gene transcription levels in the cell populations at different stages of differentiation. Individual repeat values in the histogram are indicated by black dots. Mean ± SEM, N = 3. One-way ANOVA followed by Dunnett’s post hoc test. ** *p* ≤ 0.01, *** *p* ≤ 0.001, **** *p* ≤ 0.0001.

**Figure 3 ijms-24-04871-f003:**
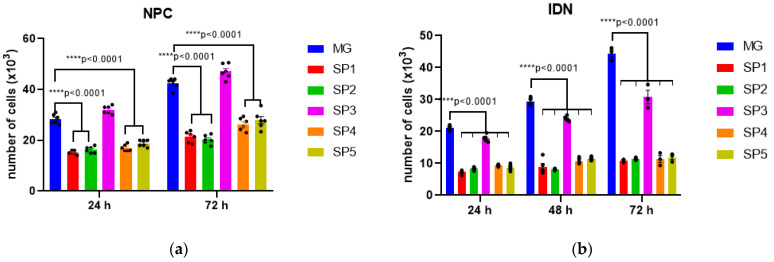
Proliferative activity of cells cultured on different CC variants 24, 48, and 72 h after seeding. (**a**) NPC, (**b**) IDN. Individual repeat values in the histogram are indicated by black dots. Mean ± SEM, N = 3–6. One-way ANOVA followed by Dunnett’s post hoc test. **** *p* < 0.0001 compared with MG.

**Figure 4 ijms-24-04871-f004:**
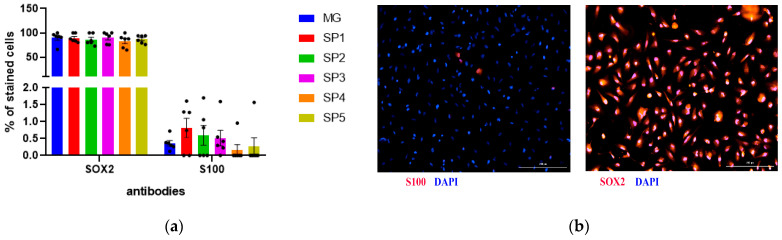
The effect of cultivation on CCs and MG on the differentiation of human NPCs. Immunocytochemical analysis of astrocyte marker S100 and neuronal marker SOX2 in human NPCs. (**a**) Percentage of S100 and SOX2 positive cells. Individual repeat values in the histogram are indicated by black dots. Mean ± SEM, N = 6. One-way ANOVA. (**b**) Immunocytochemical staining of NPCs with antibodies against S100, SOX2. Representative photographs. Scale line is 200 μm.

**Figure 5 ijms-24-04871-f005:**
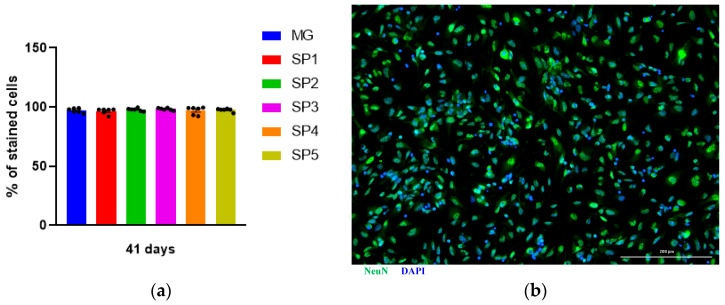
Effect of cultivation NPCs on CCs and MG on the number of NeuN-positive neurons produced. (**a**) Percentage of NeuN positive cells. Individual repeat values in the histogram are indicated by black dots. One-way ANOVA. Mean ± SEM, N = 6. (**b**) IDN immunocytochemical staining with antibodies against NeuN. Representative photographs. Scale line is 200 μm.

**Figure 6 ijms-24-04871-f006:**
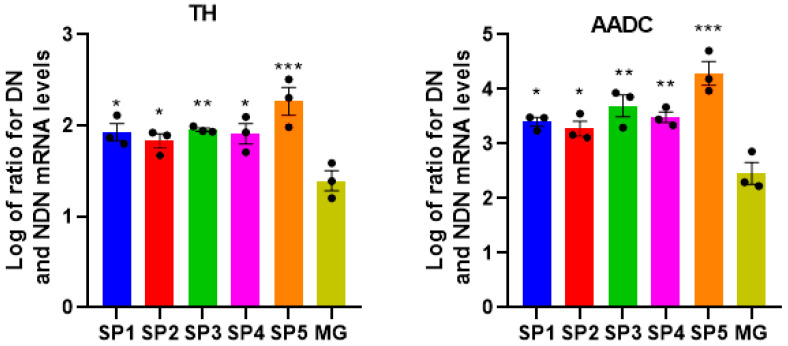
Ratios of the mRNA levels for *TH* and *AADC* between DN and NDN when NPCs were cultivated on different coatings. Individual repeat values in the histogram are indicated by black dots. Mean ± SEM, N = 3–6. One-way ANOVA followed by Dunnett’s post hoc test. * *p* ≤ 0.05, ** *p* ≤ 0.01, *** *p* ≤ 0.001 compared with MG.

**Figure 7 ijms-24-04871-f007:**
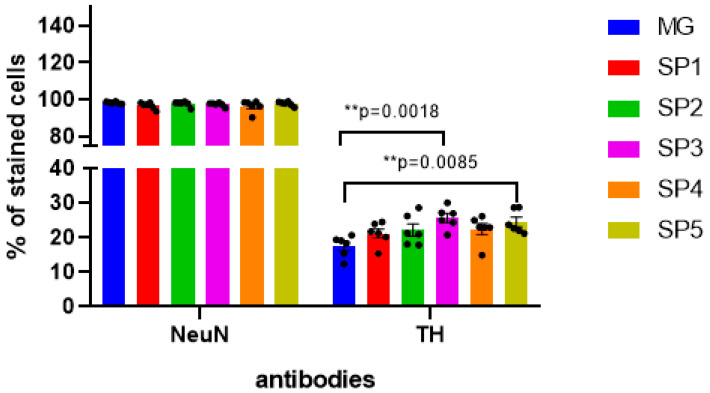
Immunocytochemical staining of DNs obtained by the cultivation of NPC on CCs and MG with antibodies against tyrosine hydroxylase (TH) and the common neuronal marker NeuN. Individual repeat values in the histogram are indicated by black dots. Mean ± SEM, N = 6. One-way ANOVA followed by Dunnett’s post hoc test. ** *p* ≤ 0.01 compared with MG.

**Figure 8 ijms-24-04871-f008:**
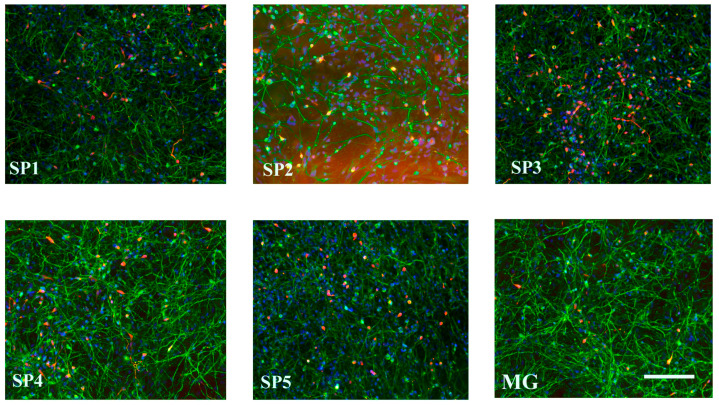
Immunocytochemical staining of DNs obtained by the cultivation of NPC on the CCs and MG using antibodies against TUBB (green), TH (red), and DAPI (blue). Representative photographs. Scale line is 100 μm.

**Figure 9 ijms-24-04871-f009:**
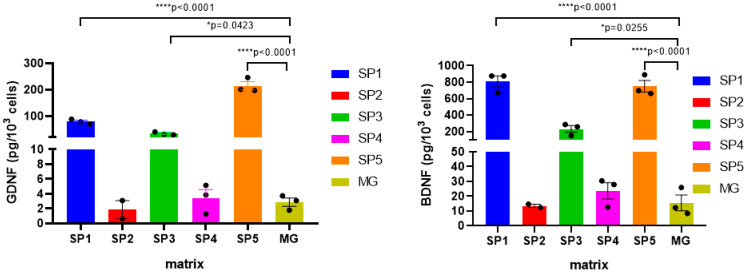
Effect of cell cultivation on the CCs on the levels of NTFs in the DNs (measured by ELISA). Individual repeat values in the histogram are indicated by black dots. Mean ± SEM, N = 3. One-way ANOVA followed by Dunnett’s post hoc test. * *p* ≤ 0.05, **** *p* ≤ 0.0001 versus MG.

**Table 1 ijms-24-04871-t001:** Effect of NPC cultivation on CCs on the expression of the neuron-specific gene.

Coating	*PAX6*	*SOX2*	*NESTIN*	*TUBB3*
SP1/MG	0.83 ± 0.22	0.83 ± 0.69	1.06 ± 0.38	0.71 ± 0.4
SP2/MG	0.96 ± 0.18	0.96 ± 0.32	1.26 ± 0.18	0.89 ± 0.16
SP3/MG	1.25 ± 0.14	1.25 ± 0.19	1.22 ± 0.14	0.78 ± 0.16
SP4/MG	1.06 ± 0.30	1.06 ± 0.42	1.27 ± 0.21	0.90 ± 0.02
SP5/MG	1.59 ± 0.25	1.59 ± 0.21	1.06 ± 0.16	0.79 ± 0.17

Presentdata on the relationships between the mRNA expression levels of the studied genes when cells were cultured on different CCs and MG.

**Table 2 ijms-24-04871-t002:** Relation of the synapse-specific gene transcription levels in NPCs cultivated on CCs compared with MG.

Coating	*SNAP 25*	*STX1A*	*SNPT*	*SYN2*	*SYN3*
	Docking of Synaptic Vesicles with the Membrane			Synaptogenesis	
SP1/MG	0.88 ± 0.14	0.62 ± 0.18	1.12 ± 0.34	0.97 ± 0.26	0.85 ± 0.34
SP2/MG	1.03 ± 0.19	0.82 ± 0.05	1.36 ± 0.34	1.30 ± 0.19	1.57 ± 0.39
SP3/MG	1.07 ± 0.18	0.94 ± 0.06	1.18 ± 0.30	1.18 ± 0.13	1.67 ± 0.35
SP4/MG	0.97 ± 0.17	0.88 ± 0.06	0.96 ± 0.04	1.08 ± 0.05	1.63 ± 0.88
SP5/MG	1.07 ± 0.17	0.81 ± 0.14	1.17 ± 0.59	1.25 ± 0.19	1.79 ± 0.69

Data on the relations of the expression levels of the studied genes in cells cultivated on CCs and MG are presented. N = 3. The Student’s t-test was used for statistical processing.

**Table 3 ijms-24-04871-t003:** Relationship between the expression levels of the synapse-specific genes in human IDNs and DNs cultivated on CCs compared with MG.

Coating	*SNAP25*		*STX1A*		*SNPT*		*PSD95*		*GSG1L*	
	Docking of Synaptic Vesicles with the Membrane						Post Synaptic Sites		Post Synaptic Density	
	IDN	DN	IDN	DN	IDN	DN	IDN	DN	IDN	DN
SP1/MG	2.07 ***	0.87	1.67 **	1.18	1.39	2.49 **	1.04	0.91	3.91 ***	1.23
SP2/MG	1.86 ***	0.93	1.40	1.12	1.26	1.71	1.05	1.07	2.48 **	1.40
SP3/MG	1.69 **	0.96	1.67 **	1.16	1.49	1.74	1.06	0.95	2.52 **	1.18
SP4/MG	0.75	1.05	1.23	1.31	1.56	0.93	0.66	0.82	1.10	1.23
SP5/MG	2.21 ***	0.78	2.63 ***	1.22	1.34	2.14 *	1.13	0.81	4.65 ***	1.38

Present data on the relationships between the expression levels of the studied genes in cells cultivated on CCs and MG. N = 3–6. One-way ANOVA followed by Dunnett’s post hoc test. * *p* ≤ 0.05, ** *p* ≤ 0.01, *** *p* ≤ 0.001 compared with MG.

**Table 4 ijms-24-04871-t004:** The ratio of the expression levels of the synapsin genes in human IDN and DN cultivated on CC compared to those cultivated on MG.

Coating	*SYN*		*SYN2*		*SYN3*	
	IDN	DN	IDN	DN	IDN	DN
SP1/MG	1.14	1.10	2.58 ***	1.03	1.31	0.75
SP2/MG	1.21	1.37	2.00 **	1.00	1.88	0.96
SP3/MG	1.14	1.21	2.08 **	1.18	2.25 *	0.83
SP4/MG	0.90	1.14	1.33	1.15	1.00	0.93
SP5/MG	1.38	1.07	3.22 ***	1.13	1.75	0.69 *

Present data on the relationships between the gene expression levels in IDNs and DNs cultivated on different CCs and MGs. N = 4. One-way ANOVA followed by Dunnett’s post hoc test. * *p* ≤ 0.05, ** *p* ≤ 0.01, *** *p* ≤ 0.001 compared with MG.

**Table 5 ijms-24-04871-t005:** Effect of various BAPs present in CCs on the expression of synapse-specific genes in human IDNs and DNs.

Coating	*SNAP 25*		*STX1A*		*SNPT*		*SYP*		*PSD95*		*GSG1L*	
	Docking of Synaptic Vesicles with the Membrane						Synaptic Protein Vesicle		Post Synaptic Sites		Post Synaptic Density	
	IDN	DN	IDN	DN	IDN	DN	IDN	DN	IDN	DN	IDN	DN
SP3/SP4	2.24 **	0.91	1.36	0.89	0.95	1.87 *	1.22	1.05	1.59	1.15 *	2.29 **	0.96
SP1/SP2	1.11	0.93	1.19	1.05	1.10	1.46	0.79	0.94	0.99	0.85	1.57 *	0.88
SP5/SP2	1.19	0.84	1.87 ***	1.09	1.06	1.25	0.80	0.83	1.08	0.75	1.87 **	0.98

The table shows the ratios of the gene expression levels on CCs with BAP versus CCs without BAP (SP2 and SP4). N = 4. The Student’s t-test was used for statistical processing. * *p* ≤ 0.05, ** *p* ≤ 0.001, *** *p* ≤ 0.0001 compared with their internal controls.

**Table 6 ijms-24-04871-t006:** Effect of the presence of various BAPs in CCs on synapsin gene expression in IDN and DN.

Coating	*SYN*		*SYN2*		*SYN3*	
	IDN	DN	IDN	DN	IDN	DN
SP3/SP4	1.27	1.06	1.50 *	1.03	2.22	0.89
SP1/SP2	0.87	0.80	1.33 *	1.02	0.68	0.78
SP5/SP2	1.11	0.78	1.78 **	1.12	0.88	0.72

The table shows the data on the ratios of the gene expression levels on CCs with BAP relative to CCs without them. N = 4. The Student’s *t*-test was used for statistical processing. * *p* ≤ 0.05, ** *p* ≤ 0.001, compared with their internal controls.

**Table 7 ijms-24-04871-t007:** Effect of cell cultivation on CCs on the levels of NTF gene transcription in the forming IDNs and DNs.

Coating	*BDNF*		*NT3*		*GDNF*		*NGF*	
	IDN	DN	IDN	DN	IDN	DN	IDN	DN
SP1/MG	3.85 ***	0.87	0.54 ***	0.77	11.92 ***	1.61	3.38 ***	1.60
SP2/MG	2.34 **	0.98	0.64 **	0.93	4.86	1.70	1.95 *	1.36
SP3/MG	3.04 **	0.83	0.61 **	0.63	5.05 *	2.06 *	2.83 ***	1.46
SP4/MG	1.33	0.85	0.50 **	0.69	3.20	1.34	1.89 *	1.12
SP5/MG	4.34 ***	0.63	0.26 ***	0.81	5.98 *	2.16 **	2.96 ***	0.98

The table shows the relations of the NTF gene expression levels in neurons cultured on CCs and MG. N = 3–4. One-way ANOVA followed by Dunnett’s post hoc test. * *p* ≤ 0.05, ** *p* ≤ 0.01, *** *p* ≤ 0.001 compared with MG.

**Table 8 ijms-24-04871-t008:** Antibodies used in the study.

Marker	Antibodies	Dilution	Company, Cat #
Primary antibodies	Mouse anti—NeuN	1:500	Abcam, #ab 104224
	Rabbit anti—TH Mouse anti—βIII Tubulin	1:1000 1:2000	Abcam, #ab 112 Abcam, #ab 7751
	Rabbit anti—Sox2	1:1000	Abcam, # 97959
	Rabbit anti—S100	1:2	Agilent Dako, #GA50461-2
Secondary antibodies	Goat anti-Rabbit IgG (H + L), AF546	1:1000	ThermoFisher, #A11010
	Goat anti- Mouse IgG (H + L), AF488	1:1000	ThermoFisher, #A11008

**Table 9 ijms-24-04871-t009:** The sequences of primers used for qPCR.

Gene	Forward	Reverse
*PAX6*	CCGAGAGTAGCGACTCCAG	CTTCCGGTCTGCCCGTTC
*SOX2*	TCCTGATTCCAGTTTGCCTC	GCTTAGCCTCGTCGATGAAC
*NESTIN*	CAGCTGGCGCACCTCAAGATG	AGGGAAGTTGGGCTCAGGACTGG
*TUBB3*	CTCAGGGGCCTTTGGACATC	CAGGCAGTCGCAGTTTTCAC
*TH*	GTCCCCTGGTTCCCAAGAAAAGT	TCCAGCTGGGGGATATTGTCTTC
*AADC*	CTCGGACCAAAGTGATCCAT	GGGTGGCAACCATAAAGAAA
*BDNF*	ATTGGCTGGCGATTCATAAG	GTTTCCCTTCTGGTCATGGA
*NT3*	AACTGCTGCGACAACAGAGA	CCAGCCCACGAGTTTATTGT
*TRKC*	CCGACACTGTGGTCATTGGCAT	CAGTTCTCGCTTCAGCACGATG
*GDNF*	ACCTGGAGTTAATGTCCAACC	GGCATATTTGAGTCACTGCT
*GFRα*	GCCTGTGTGCTCCTATGAAG	CTGGCTGGCAGTTGGTAAA
*NGF*	TAAAAAGCGGCGACTCCGTT	ATTCGCCCCTGTGGAAGATG
*P75NTR*	AAGGAGGGGAGGTGATAGGA	GTGGGACAGAGTCTGGGTGT
*SNAP25*	CGTCGTATGCTGCAACTGGTTG	GGTTCATGCCTTCTTCGACACG
*STX1A*	TGGAGAACAGCATCCGTGAGCT	CCTCTCCACATAGTCTACCGCG
*SNPT1*	GCTGACTGTTGTCATTCTGGAGG	CTTCAGCCTCTTACCATTCTGCA
*SYP*	TCGGCTTTGTGAAGGTGCTGCA	TCACTCTCGGTCTTGTTGGCAC
*PSD95*	TCCACTCTGACAGTGAGACCGA	CGTCACTGTCTCGTAGCTCAGA
*SYN3*	GTGGACATGCAGGTCGTGAGAA	ATGACCAGGCTGCGGTAGTCTT
*18S*	CGGCTACCACATCCAAGGAA	GCTGGAATTACCGCGGCT
*GSG1L*	CTCCTACACCAAGACGGTCATTG	GGCAGTCTAAGTGAAAGTCCTCC
*SYN*	CGATGCCAAATATGACGTGCGTG	AGCATCGCAGAGCCAGTATTGG
*SYN2*	ACCTGCTCTGAGATGTTTGGCG	GTTCGGTGATGAGTTGCCTGTC

## Data Availability

The datasets used and/or analyzed during the current study are available from the corresponding author on reasonable request.

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
