# Peer review of "Composite Coatings Based on Recombinant Spidroins and Peptides with Motifs of the Extracellular Matrix Proteins Enhance Neuronal Differentiation of Neural Precursor Cells Derived from Human Induced Pluripotent Stem Cells"

_ijms, 2023, doi:10.3390/ijms24054871_

Round 1

Reviewer 1 Report

In the manuscript titled “Composite Coatings Based on Recombinant Spidroins and Pep-tides with Motifs of the Extracellular Matrix Proteins Enhance Neu-Ronal Differentiation of Neural Precursor Cells Derived from Human Induced Pluripotent Stem Cells”, authors have evaluated the suitability of novel composite coatings containing recombinant spidroins rS1/9 and rS2/12 in combination with recombinant fused proteins carrying bioactive motifs of the extracellular matrix proteins, for growth of NPCs derived from human induced pluripotent stem cells and their differentiation into neurons and found that a mixture of two RSs and FPs with different peptide motifs of ECMs in-creases the efficiency of obtaining DNs differentiated from iPSCs compared to Matrigel. This approach might promote the maturation of neurons and achieve the adult like characteristics in differentiated and mature neuron and also accelerate recapitulation of pathological events in iPSC derived neurons from patient , that iPSC derived neurons lack.

Despite the novelty, manuscript requires minor revision.

1.       How closely are DNs mimicking the adult state?

2.       Do novel CCs degrade over the time as MG does?

3.       Could you do immunostaining on DNs for the neuronal markers to compare the efficacy of CCs?

4.       In some bar graphs, the color of bars is hardly distinguishable, e.g., in figure 2.  

5.       Adding individual values of repeats in bar graphs would be more valuable. 

Reviewer 2 Report

Overview of the manuscript
The manuscript focuses on the studying particular matrices to promote the growth and differentiation of neural precursor cells (NPCs) into the desired neuronal types. In particular the authors want to evaluate suitability of novel composite coatings (CCs) containing re-combinant spidroins (RS) rS1/9 and rS2/12 in combination with recombinant fused proteins (FP) carrying bioactive motifs (BAP) of the extracellular matrix (ECM) proteins. NPCs were produced by directed differentiation of human iPSCs. The authors observed that CC consisting of two RS and FP with RGDS and HBP is the most effective for the support of NPCs and their neuronal differentiation.

GENERAL COMMENT

The work is very interesting. The use of composite coating well illustrates the importance of a mix of artificial adhesive component and peptide motif of structural matrix proteins.

The results are obtained through a rich and rigorous methodological approach, giving solidity to the conclusion of the work. Following some suggestions to improve the reading of the work.

SPECIFIC COMMENTS

 Abstract

Pag. 1, line 14-15: explicit the acronyms

Introduction

Pag. 2, line 13: insert references

Results

Insert a visualization of some blot in results.

 Discussion

Please provide more detailed comments on the differences observed between the several composite coating investigated
